# A cVLP-Based Vaccine Displaying Full-Length PCSK9 Elicits a Higher Reduction in Plasma PCSK9 Than Similar Peptide-Based cVLP Vaccines

**DOI:** 10.3390/vaccines11010002

**Published:** 2022-12-20

**Authors:** Louise Goksøyr, Magdalena Skrzypczak, Maureen Sampson, Morten A. Nielsen, Ali Salanti, Thor G. Theander, Alan T. Remaley, Willem A. De Jongh, Adam F. Sander

**Affiliations:** 1Centre for Medical Parasitology at Department for Immunology and Microbiology, Faculty of Health and Medical Sciences, University of Copenhagen, 2200 Copenhagen, Denmark; 2AdaptVac Aps, 2200 Copenhagen, Denmark; 3ExpreS2ion Biotechnologies Aps, 2970 Hørsholm, Denmark; 4Lipoprotein Metabolism Section, Translational Vascular Medicine Branch, National Heart, Lung and Blood Institute, National Institutes of Health, Bethesda, MD 20892, USA

**Keywords:** vaccine, virus-like particle (VLP), PCSK9, immune complex, cholesterol-lowering vaccine, SpyTag, SpyCatcher

## Abstract

Administration of PCSK9-specific monoclonal antibodies, as well as peptide-based PCSK9 vaccines, can lower plasma LDL cholesterol by blocking PCSK9. However, these treatments also cause an increase in plasma PCSK9 levels, presumably due to the formation of immune complexes. Here, we utilize a versatile capsid virus-like particle (cVLP)-based vaccine platform to deliver both full-length (FL) PCSK9 and PCSK9-derived peptide antigens, to investigate whether induction of a broader polyclonal anti-PCSK9 antibody response would mediate more efficient clearance of plasma PCSK9. This head-to-head immunization study reveals a significantly increased capacity of the FL PCSK9 cVLP vaccine to opsonize and clear plasma PCSK9. These findings may have implications for the design of PCSK9 and other vaccines that should effectively mediate opsonization and immune clearance of target antigens.

## 1. Introduction

Development of capsid virus-like particle (cVLP)-based vaccines displaying large and complex protein antigens has long been technically challenging [1,2], but has been made possible by the use of split-protein conjugation technology that can facilitate covalent attachment of antigens to the surface of pre-assembled cVLPs [3,4,5,6]. Specifically, genetic fusion of a split-protein binding partner (i.e., SpyTag (SpyT) or SpyCatcher (SpyC)) to the *Acinetobacter phage* AP205 capsid protein has enabled the formation of cVLPs, each displaying a total of 180 SpyT or SpyC binding partners. When mixed, SpyT and SpyC react to form a covalent binding through an isopeptide bond [6]. Accordingly, unidirectional conjugation of antigen to the cVLP surface is achieved simply by mixing antigen and cVLP, each genetically fused to the complementary binding partner [2,3,4,6]. The resultant high-density epitope display significantly increases the immunogenicity of the antigen and has been shown to overcome B-cell immune tolerance and induce strong autoantibody responses against disease-associated self-antigen targets [3,7,8,9,10,11]. Thus, this vaccine technology allows for unique head-to-head comparisons of various antigen designs, ranging in both size and complexity [2,12]. Exploiting this, we aimed to investigate whether antibody responses induced by cVLP vaccines displaying either a full-length (FL) antigen or antigen-derived peptides would differ in their capacity to neutralize a self-antigen vaccine target. For this purpose, we chose proprotein convertase subtilisin/kexin type 9 (PCSK9) as an interesting model antigen.

PCSK9 is produced in the liver, where it acts as a negative regulator of the low-density lipoprotein receptor (LDLR) by preventing the recycling of the receptor to the cell surface [13,14]. Thus, binding of PCSK9 causes internalization and degradation of the LDLR, thereby increasing the levels of circulating LDL cholesterol (LDL-C) and, in turn, the risk of cardiovascular disease (CVD) [15]. Humans with no detectable circulating PCSK9 have exceptionally low levels of serum LDL-C (<20 mg/dL), with no apparent associated health issues [16,17]. Accordingly, people with loss-of-function mutations in PCSK9 have an approximately 50% lower risk of CVD, likely due to having a life-long reduction in LDL-C levels [14,16,18,19,20,21,22]. This has made PCSK9 a promising therapeutic target in relation to maintaining cholesterol homeostasis and treatment of atherosclerotic CVD.

Since 2015, PCSK9-specific monoclonal antibodies (mAb) (i.e., Evolocumab (Amgen) and Alirocumab (Regeneron)) have been approved by the U.S. Food and Drug Administration (FDA) and European Medicines Agency (EMA) as second-line treatments for patients with high LDL-C levels [23,24,25,26,27,28]. These mAbs act in synergy with statins to lower plasma LDL-C levels by approximately 60% [29]. However, there are many limitations to mAb therapy, both in terms of patient access and clinical utility. The required frequent administration of high doses of mAb is expensive, and repeated administrations of mAb can lead to the induction of anti-drug immunity followed by loss of drug efficacy [30,31,32]. Moreover, elevated levels of plasma PCSK9 have been reported following the administration of anti-PCSK9 mAbs, indicating that anti-PCSK9 mAbs prevent PCSK9 from interacting with the LDLR by forming immune complexes with a decreased plasma clearance rate [33]. Similar results have been reported from studies on pre-clinical PCSK9 vaccines using shorter PCSK9–peptide antigens. Here, induction of anti-PCSK9 antibody responses, targeting known neutralizing B-cell epitopes, has been shown to decrease levels of total cholesterol (TC) and LDL-C in mice, rats and macaques [34,35,36,37,38,39,40,41,42]. However, the majority of these studies measured significantly elevated total plasma PCSK9 levels [34,35,36,37,38,41]. As plasma PCSK9 levels could be considerably reduced by IgG depletion [34,35], it is presumed that these anti-PCSK9 antibodies do not opsonize and clear serum PCSK9, but instead are forming immune complexes, as observed for mAb therapy [33,34,35]. Thus, we believe that PCSK9 serves as an interesting target to compare the neutralizing capacity of vaccine-induced antibody responses induced by peptide versus FL antigen designs, respectively. Consequently, we developed an FL murine (mu) PCSK9 cVLP vaccine, as well as two peptide-based muPCSK9 cVLP vaccines, which we tested head to head. Specifically, the immunogenicity of the PCSK9-cVLP vaccines was assessed in mice, and the biological effects of vaccine-induced antibodies were investigated in vivo by measuring changes in plasma TC, triglycerides (TGs) and PCSK9 levels. The obtained data show a significant difference in the capacity of the FL and peptide-based PCSK9 cVLP vaccines to opsonize and clear plasma PCSK9.

## 2. Materials and Methods

### 2.1. Design, Expression and Purification of Spy-cVLP

cVLPs assembled from the major coat protein of The *Acinetobacter phage*, AP205 (Gene ID: 956335), genetically fused to either SpyTag or SpyCatcher (Genbank: OK545878.1 and OK422508.1, respectively) was designed and produced, as previously described [3]. Purified Spy-cVLPs were dialyzed into PBS, pH 7.4 using cutoff 1000 kDa (SpectrumLabs, San Francisco, CA, USA). Protein concentration was determined by the Bicinchoninic acid (BCA) assay (Thermo Fisher Scientific, Waltham, MA, USA).

### 2.2. Design, Expression and Purification of Recombinant muPCSK9 Antigens

FL murine PCSK9 comprising amino acid 35–694 (GenPept: NP_705793) was designed with a C-terminal 6xHis-tag and SpyT, separated by a flexible Gly-Gly-Ser linker. The small SpyT was chosen as the binding partner to minimize the impact of the protein engineering and the subsequent risk of protein misfolding. Flanking EcoRI and NotI restriction sites were added to the N- and C-terminus, respectively. The gene sequence was codon-optimized for expression in *Drosophila melanogaster* and synthesized by Geneart©. To prevent furin cleavage during expression, the mutation R221S was introduced by overlap-extension PCR and verified by sequencing. Primers used for PCR are listed in Appendix A. The final gene sequence was cloned into the pExpres2-1 vector (ExpreS^2^ion Biotechnologies, Hørsholm, Denmark). The ExpreS^2^ expression platform was used to obtain a stable cell line producing the FL PCSK9-SpyT antigen. In brief, Schneider-2 (ExpreS^2^) cells were transfected using ExpreS^2^ Insect TRx5 transfection reagent (ExpreS^2^ion Biotechnologies) according to the manufacturer’s protocol and transfected cells were selected with zeocin (2 mg/mL, Thermo Fisher Zeocin™ Selection Reagent, R25005). After selection, the cells were expanded by culturing in shake flasks at 25 °C with handling every 3–4 days. The supernatant was harvested via centrifugation at 5000 rpm (6227 g) for 10 min at 4 °C (Beckman Avanti JVN-26 centrifuge using a JLA 8.1000 rotor). The supernatant was filtered through a 0.22 µm vacuum filter (PES) and frozen before further purification. The supernatant was thawed and filtered (0.2 µm), before up-concentration and buffer exchanged into PBS, pH 7.4 using a Quixstand (Hollow Fiber cartridge cutoff 10 kDa, GE Healthcare, Chicago, IL, USA). Recombinant FL PCSK9-SpyT was purified by Immobilized metal chelate affinity (IMAC) using a 5 mL HisTrap™ HP affinity column (GE Healthcare). The column was washed with binding buffer (125 mM Tris-HCl, 500 mM NaCl, 0.25 mM CaCl_2_, 5% glycerol, 60 mM Imidazole, pH7.2). The protein was eluted in 125 mM Tris-HCl, 500 mM NaCl, 0.25 mM CaCl_2_, 5% glycerol, 500 mM Imidazole, pH 7.2. Further purification was performed by size-exclusion chromatography (SEC) using a HiLoad Superdex 200 pg column (GE Healthcare) (125 mM Tris-HCl, 500 mM NaCl, 0.25 mM CaCl_2_, 5% glycerol, pH 7.2).

Two peptide PCSK9 antigens (muPCSK9 (Gene ID: 100102) amino acids 210–226 and 156–227) were designed with an N-terminal 6xHis-tag and SpyC. A flexible Gly-Gly-Ser linker was inserted between the SpyC and muPCSK9 sequence. The larger SpyC binding partner was chosen for the peptide-based antigens to serve as a protein chaperone during expression, as well as to ease the purification process. Flanking NcoI and NotI restriction sites were added to the N- and C-terminus, respectively. Primers used for PCR are listed in Appendix A. The final gene sequences were cloned into the pET15b vector. Plasmids were transformed into *E. coli* OneShot^®^ BL21 Star™ (DE3) cells (Thermo Scientific). Recombinant SpyC-muPCSK9 constructs were purified by IMAC (PBS, pH 7.4 with either 60 mM or 500 mM imidazole for binding and elution, respectively) and SEC using HiLoad Superdex 75 pg column (GE Healthcare) (PBS, pH 7.4).

### 2.3. Vaccine Formulation

Triton X-114 was used for the removal of endotoxins from both cVLPs and PCSK9 antigens before vaccine formulation, as described in [43]. To formulate the cVLP-PCSK9(FL) vaccine, the FL PCSK9-SpyT antigen was dialyzed into 100 mM HEPES, 500 mM NaCl, pH 7 overnight (O/N) at 4 °C and subsequently mixed in a 1:1.2 molar ratio (cVLP subunit per antigen) with the addition of HEPES and L-Arginine to a final concentration of 250 mM and 500 mM, respectively. The vaccine was incubated O/N at 4 °C and purified on an Optiprep™ (Sigma-Aldrich, St. Louis, MO, USA) density gradient (23, 29 and 35%) by ultracentrifugation, as described previously [3]. The final vaccine was dialyzed into 250 mM HEPES, 500 mM NaCl, 500 mM L-Arginine, pH 7 O/N at 4 °C (1000 MWCO, SpectrumLabs). Formulation of the peptide PCSK9 cVLP vaccines (cVLP-PCSK9(210–226) and cVLP-PCSK9(156–227)) was performed by mixing SpyT-cVLP and SpyC-PCSK9 antigen in a 1:2 molar ratio in PBS, pH 7.4, followed by incubation O/N at 4 °C. The vaccines were purified by ultracentrifugation and dialyzed (PBS, pH 7.4) O/N at 4 °C (1000 MWCO, SpectrumLabs), as described above. All cVLP-PCSK9 vaccines were subjected to a centrifugation stability test (16,000 g for 2 min). Pre- and post-spin samples were loaded on a reduced SDS-PAGE (Nu-PAGE 4–12% Bis-Tris polyacrylamide gel (Invitrogen, Waltham, MA, USA)) to evaluate potential precipitation, indicating vaccine aggregation. Antigen concentration on the cVLP and coupling efficiency were estimated using densitometry from SDS-PAGE gels (Nu-PAGE 4–12% Bis-Tris polyacrylamide gel) using ImageQuant TL (Cytiva, Marlborough, MA, USA). Coupling efficiency was calculated as (coupled cVLP subunits/coupled cVLP subunits + uncoupled cVLP subunits).

### 2.4. Quality Assessment of cVLP-PCSK9 Vaccines

Purified PCSK9 vaccines were quality assessed by DLS and negative-stain TEM. For DLS analysis, vaccines were diluted to ~0.5 mg/mL and spun at 16,000 g for 2 min before being loaded into a disposable cuvette. The samples were run with 20 acquisitions of 7 s, at 25 °C using a DynaPro Nanostar (Wyatt Technologies, Santa Barbara, CA, USA). The estimated diameter of the cVLP-PCSK9 vaccine population and percent polydispersity (%Pd) were calculated by Wyatt DYNAMICS software (v7.10.0.21, US) (Wyatt Technologies). For TEM, vaccines were adsorbed onto 200-mesh carbon-coated grids and stained with 2% uranyl acetate. The grids were analyzed using a CM 100 BioTWIN electron microscope (Philips, Amsterdam, Netherlands), with an accelerating voltage of 80 kV.

### 2.5. Immunofluorescence Staining of Hepa1-6 Cells

Murine Hepa1-6 hepatocytes were a kind gift from the Department of Immunology and Microbiology, Copenhagen Hepatitis C program (CO-HEP). Hepa1-6 cells were cultured in DMEM High-Glucose (Sigma-Aldrich), 10% heat-inactivated fetal bovine serum (FBS) (Gibco), 1% L-glutamine, 100 U/mL penicillin and 100 µg/mL streptomycin.

Hepa1-6 cells were seeded into a 24-well cell culture plate, containing 13 mm glass slides (VWR, Radnor, PA, USA) at 10,000 cells/cm^2^ (19,000 cells/well) and incubated for 24 h. Following 16 h of serum deprivation, cells were incubated with 100 nM muPCSK9 (i.e., recombinant control FL muPCSK9 (Abcam, Cambridge, UK) or FL PCSK9-SpyT antigen) or growth media without FBS for 4 h at 37 °C. Cells were washed in cold PBS and fixed with 4% paraformaldehyde for 10 min. Cells were washed three times in PBS before the glass slides containing the fixed cells were moved from the 24-well plate to a “staining humidity chamber”, where the following staining and washing steps were performed. All slides containing cells were permeabilized with PBS containing 0.2% Triton X-100 for 10 min and washed three times in PBS. Cells were blocked (2% BSA, 0.05% Triton X-100) for 1 h at RT followed by incubation with goat anti-mouse LDLR polyclonal antibody (5 µg/mL, Invivogen, San Diego, CA, USA) in PBS, 0.1% BSA, 0.05% Triton X-100 for 1 h at RT. Cells were washed three times in PBS and stained with donkey anti-goat IgG-FITC (Invitrogen) (1:300 in PBS, 0.1% BSA, 0.05% Triton X-100) for 45 min at RT. Cells were washed three times in PBS, with a DAPI stain (Invitrogen) included in the second wash (1:1000, incubated for 5 min at RT). Slides were washed once in UPW and left to dry. Slides were mounted upside-down on larger glass slides, before analysis by Cytation5 (Agilent Technologies, Santa Clara, CA, USA).

### 2.6. Immunization Study

Animal experiments were authorized by the Danish National Animal Experiments Inspectorate (Dyreforsøgstilsynet, license no. 2018-15-0201-01541) and performed according to national guidelines. Mice were fed on the SAFE^®^ D30 diet. Thus, 6–8-week-old male BALB/c mice (Janvier, Denmark) were immunized intramuscularly (IM) in a 3-week interval prime-boost-boost regime. Specifically, mice received (1) 2.4 µg (364 pmol) cVLP-displayed muPCSK9(210–226), (2) 3 µg (364 pmol) cVLP-displayed muPCSK9(156–227), (3) 7.3 µg (99 pmol) cVLP-displayed muPCSK9(35–694) R221S, (4) 10 µg soluble muPCSK9(35–694)R221S-SpyT or (5) 5 µg SpyC-cVLP. As the size of the PCSK9 antigens varies greatly, the dose was based on molar concentrations. Due to limited protein concentration, the maximum dose for the cVLP-PCSK9 FL vaccine was approximately a third of the peptide-based vaccines. The dose of the soluble PCSK9-SpyT was administered at a higher dose to demonstrate that the absence of a PCSK9-specific Ab response in this group was not due to a low dose, but the absence of the cVLP display. The displayed antigen concentration on the cVLP was calculated by densitometric measurement (ImageQuant TL), using a reference protein at a known concentration and a size close to the respective protein coupling bands. All vaccines were formulated 1:1 in Addavax^TM^ (Invivogen). Blood samples were taken in the morning prior to the first immunization (pre-bleed) as well as two weeks after each immunization in EDTA tubes. Plasma was obtained by spinning the blood twice for 10 min at 2000 g, 8 °C. On the day of termination, a full bleed was taken.

### 2.7. Analysis of Vaccine-Induced Antibody Responses

PCSK9-specific total IgG and subclass titers were measured by ELISA. First, 96-well plates (Nunc MaxiSorp, Invitrogen) were coated O/N at 4 °C with 0.1 µg/well muPCSK9 (Abcam) or muPCSK9(35–694) R221S-SpyT in PBS, pH 7.4. Plates were blocked with 0.5% skimmed milk in PBS (i.e., blocking buffer) O/N at 4 °C. Mouse plasma diluted in a 3-fold dilution starting from 1:50 was added to the plate followed by incubation for 1 h at RT. Plates were washed 3 times in PBS in between steps. Total plasma IgG was detected using Horseradish peroxidase (HRP)-conjugated goat-anti mouse IgG (Invitrogen, A16072) diluted 1:1000 in blocking buffer and incubated for 1 h at RT. To measure IgG subclass, HRP goat anti-mouse IgG1 (Invitrogen, A10551), HRP goat-anti mouse IgG2a (Invitrogen, M32207), HRP goat-anti mouse IgG2b (Invitrogen, M32407) and HRP goat-anti mouse IgG3 (Thermo Ficher, M32707) were diluted 1:1000 in blocking buffer and incubated for 1 h at RT. Plates were developed with TMB X-tra substrate (Kem-En-Tec, Taastrup, Denmark) and the reaction was stopped with 0.2 mM H_2_SO_4_. The absorbance was measured at 450 nm using a BioSan HiPo MPP-96 microplate reader (BioSan, Riga, Latvia).

### 2.8. Plasma PCSK9 Levels

Plasma PCSK9 levels (i.e., total PCSK9, free PCSK9 and IgG-bound PCSK9) were measured by a quantitative mouse PCSK9 ELISA assay (R&D systems, Minneapolis, MN, USA), by comparing plasma from vaccinated mice and pre-bleeds to a standard curve. Pre-bleed was not collected for each individual mouse and, thus, it was not possible to show the individual change in plasma PCSK9 levels. For total PCSK9, all plasma samples were diluted 1:200. The assay was run according to the manufacturer’s protocol.

Plasma from each vaccination group was pooled and IgG purified using GammaBind™ Plus Sepharose™ (GE Healthcare). Plasma samples were diluted 1:1 in 20 mM Na_3_PO_4_, pH 7 before being loaded on the column. The run-through was collected and loaded on the column five times to ensure the depletion of all IgG. Following a column wash (100 mM citric acid, pH 4), IgG was eluted using 100 mM citric acid, pH 2.7 into a neutralization buffer (1 M Tris-HCl, pH9) to a final pH of 7. Purified IgG was up-concentrated and buffer exchanged into PBS pH 7.4 by Vivaspin 500 (30 kDa cutoff). For PCSK9 quantitative ELISA, the run through was further diluted 1:100, while 10 µg purified IgG was used. IgG concentration was measured by Nanodrop.

### 2.9. Plasma Lipid Profile

Plasma lipids were measured enzymatically using a Chemwell instrument (Awareness Technology, Palm City, FL, USA), using Wako reagents, cholesterol oxidase for TC and glycerol-blanked GPO Trinder for TG measurements.

### 2.10. Statistical Analysis

Statistical analysis was performed on log-transformed values using one-way ANOVA Tukey’s multiple comparisons test (adjusted *p*-value < 0.05 was accepted as significant). Statistical analysis was performed using GraphPad Prism (9.3.1) (GraphPad, San Diego, CA, USA).

## 3. Results

### 3.1. Development and Characterization of cVLP-Based PCSK9 Vaccines

To investigate the biological effect of vaccine-induced anti-PCSK9 antibody responses, we developed three vaccines based on FL protein or peptides representing muPCSK9 antigens, delivered by the same SpyT/SpyC AP205 cVLP vaccine platform [3]. The first peptide antigen was based on aa210–226 of the murine PCSK9 sequence (i.e., corresponding to aa207–223 of human PCSK9), which has previously been shown to induce anti-PCSK9 antibodies with the capacity for lowering TC and TG in mice [34]. The second peptide antigen comprised aa156–227, which covers additional B-cell epitopes included in other peptide-based PCSK9 vaccines [34,35]. Both of the short antigen constructs were N-terminally fused by a flexible linker to SpyC (Figure 1A) and expressed as recombinant proteins in *E. coli*. The third antigen comprised the FL muPCSK9 aa35–694, with an introduction of an R221S mutation to prevent furin cleavage during recombinant protein production in S2 insect cells [44,45]. The FL construct was C-terminally fused to SpyT and separated by a flexible linker (Figure 1A). For cVLP vaccine formulation, the SpyC-PCSK9 peptide antigens were mixed with AP205 cVLPs displaying SpyT (SpyT-cVLP) in a 2:1 molar ratio (Figure 1B, top), whereas the FL PCSK9-SpyT antigen was mixed in a 1.2:1 molar ratio with AP205 cVLP displaying SpyC (SpyC-cVLP) (Figure 1B, bottom). Upon mixing, the SpyT and SpyC binding partners react to form a covalent bond, thus, enabling high-density, unidirectional antigen display on the cVLP surface (Figure 2B).

Covalent coupling of PCSK9 antigens to the cVLP was confirmed by SDS-PAGE analysis by the appearance of a protein band with a size corresponding to the conjugated product of the cVLP subunit and the PCSK9 antigen. Thus, a protein band of 32 kDa and 38.5 kDa could be seen after conjugation of the two peptide antigens (i.e., SpyC-PCSK9(210–226) (15.5 kDa) and SpyC-PCSK9(156–227) (22 kDa)) to the SpyT-AP205 cVLP (16.5 kDa), respectively (Figure 2A, lane 3 and 5). FL PCSK9-SpyT (73 kDa) consists of a pro-domain (14 kDa) non-covalently bound to the catalytic domain and a C-terminal domain (59 kDa). Consequently, the protein band occurring after conjugation of the FL PCSK9 antigen to SpyC-cVLP (27 kDa) was seen around 86 kDa, as it excludes the pro-domain (Figure 2E, lane 2). To assess vaccine stability, a centrifugation test (16,000 g, 2 min) was conducted. In this case, centrifugation of the individual vaccines led to no detectable loss in the protein coupling band, indicating that vaccine formulations were stable (Figure 2A, lane 3–6, and Figure 2E, lane 2–3). Antigen coupling efficiencies (i.e., the percentage of binding sites per cVLP conjugated to an antigen) were estimated to be ~80%, ~60% and ~40% for the cVLP-PCSK9(210–226), cVLP-PCSK9(156–227) and cVLP-PCSK9(FL) vaccine, respectively (Figure 2A,E). The cVLP PCSK9 vaccines were further analyzed by Dynamic Light Scattering (DLS) and Transmission electron microscopy (TEM). DLS analysis showed a multimodal sample for the two peptide-based cVLP-PCSK9 vaccines, shown by the presence of two peaks (Figure 2B). The bigger particle size (i.e., >70 nm) and higher polydispersity (%Pd) for the main peak (i.e., peak 2) indicate particles interacting with each other, supported by TEM (Figure 2B–D). By contrast, the cVLP-PCSK9(FL) vaccine showed a clear monodisperse sample of ~80 nm (Figure 2F). The presence of intact cVLP-PCSK9(FL) particles was confirmed by TEM (Figure 2G,H).

### 3.2. Biological Activity of FL PCSK9-SpyT

To further assess the quality of the recombinant FL PCSK9-SpyT antigen, the biological activity was investigated in a cell-based assay. As PCSK9 acts as a negative regulator of LDLR, incubation of hepatocytes with PCSK9 is expected to lower the level of surface-expressed LDLR [13,14]. Accordingly, mouse Hepa1-6 hepatocytes were incubated with either media (control), recombinant FL PCSK9 or the FL PCSK9-SpyT antigen, followed by staining of the LDLR. Immunofluorescence staining showed significantly lower levels of the LDLR on cells incubated with 100 nM FL PCSK9 or FL PCSK9-SpyT, compared to control (*p* < 0.0001) (Figure 3). Importantly, recombinant FL PCSK9 and FL PCSK9-SpyT reduce the LDLR level to a similar degree (Figure 3B), verifying the fold and biological activity of the FL PCSK9-SpyT antigen. A reduction in the LDLR level was also observed following incubation with the cVLP-PCSK9 FL vaccine, demonstrating that the antigen retains its biological activity following cVLP display (Appendix A).

### 3.3. Immunogenicity of cVLP-PCSK9 Vaccines

The immunogenicity of the cVLP-PCSK9 vaccines was assessed in plasma from male BALB/c mice vaccinated in a three-week interval prime-boost-boost regimen. Mice received similar molar antigen doses of PCSK9(210–226) or PCSK9(156–227) (i.e., 364 pmol). FL PCSK9 was administered at a lower molar antigen dose (i.e., 99 pmol) or as a soluble antigen. All vaccines were formulated in squalene–water emulsion adjuvant (Addavax™). Antigen-specific IgG titers were measured by ELISA, using FL PCSK9-SpyT for capture. The capture protein was tested head-to-head with a commercial FL PCSK9 protein, showing no significant difference (Appendix A). All cVLP-PCSK9 vaccines led to seroconversion in all mice (Figure 4B). Mice vaccinated with the peptide cVLP-PCSK9 vaccines (i.e., cVLP-PCSK9(210–226) or cVLP-PCSK9(156–227)) had significantly higher final total anti-PCSK9 IgG levels than mice vaccinated with the cVLP-PCSK9(FL) vaccine (*p* = 0.0133 and *p* < 0.0001, respectively) or soluble FL PCSK9-SpyT (Figure 4A,B). There was no significant difference in total anti-PCSK9 antibody levels measured in plasma from mice immunized with soluble or cVLP-displayed FL PCSK9, nor between the two peptide-based cVLP-PCSK9 vaccines (Figure 4A,B). For the two peptide-based vaccines, the peak response was obtained already after the first boost immunization (i.e., 2nd bleed), whereas the second boost immunization had a significant effect on the cVLP-PCSK9(FL) response (*p* < 0.0001) (Figure 4B). The IgG subclass profile was assessed for all vaccine groups (Figure 4C–F). Mice vaccinated with soluble PCSK9-SpyT responded almost exclusively with IgG1 antibodies (Figure 4C–F), whereas all the cVLP-PCSK9 vaccines also induced IgG2a/b, indicating a more balanced Th1/Th2-type immune response (Figure 4C–F). Finally, both peptide-based cVLP-PCSK9 vaccines induced IgG3 antibodies, whereas these could not be measured in mice vaccinated with cVLP-PCSK9(FL) (Figure 4E).

### 3.4. Immunization with cVLP-PCSK9 Vaccines Decreases Total Cholesterol Levels in Mice

To evaluate the biological effect of the vaccine-induced antibodies, the TC and TG levels were measured in plasma from vaccinated mice. For all the cVLP-PCSK9 vaccine groups, TC levels were significantly reduced in post-vaccination plasma samples compared to the levels measured in the pre-vaccination plasma samples. A significant reduction in TC (19.6%) was observed in mice vaccinated with cVLP-PCSK9(FL) (*p* < 0.0001) (Figure 5A). In comparison, mice vaccinated with cVLP-PCSK9(210–226) and cVLP-PCSK9(156–227) showed a 16.2% (*p* = 0.0006) and 14.4% (*p* = 0.0053) reduction in TC, respectively (Figure 5A). Moreover, cVLP display of FL PCSK9-SpyT results in a statistically significant reduction in TC (11.6%), as compared to mice vaccinated with soluble FL PCSK9-SpyT (*p* = 0.0002). We found no significant difference in the TC reduction level between the cVLP-PCSK9 vaccines. Additionally, we found no statistically significant decrease in TC levels for mice vaccinated with soluble FL PCSK9-SpyT. Furthermore, there was observed no statistically significant decrease in TG levels, in any of the cVLP-PCSK9 vaccinated groups.

### 3.5. cVLP-PCSK9(FL) Vaccine Decreases Plasma PCSK9 Levels

The biological effect of the vaccine-induced antibodies was further assessed by measuring the quantitative PCSK9 level in plasma from immunized mice. This analysis showed a significant difference in the total PCSK9 level between the cVLP-PCSK9-vaccinated groups (Figure 5C). The total plasma PCSK9 level was significantly lower in mice vaccinated with cVLP-PCSK9(FL) than in mice vaccinated with either of the peptide PCSK9 cVLP vaccines (*p* < 0.0001; cVLP-PCSK9(210–226) and *p* = 0.0002; cVLP-PCSK9(156-227)). Although, there was no statistically significant difference between the total PCSK9 level in pre-vaccination (pre-bleed) plasma and plasma obtained after immunization with either of the cVLP-PCSK9 vaccines, there was a clear trend that PCSK9 levels were higher in mice vaccinated with either of the peptide-based PCSK9 cVLP vaccines (Figure 5C). In fact, the level of total PCSK9 quantified in mice vaccinated with the peptide-based PCSK9 cVLP vaccines was statistically significantly higher than in mice vaccinated with cVLP-PCSK9 FL. To investigate the presence of antibody-PCSK9 circulating immune complexes, pooled plasma from vaccinated mice was IgG purified to obtain an IgG-free fraction and an IgG-containing fraction. The PCSK9 level was quantified in each fraction, to gain information on the relative proportion of free and IgG-bound PCSK9 in plasma from vaccinated mice. Here, mice immunized with cVLP-PCSK9(210–226) and cVLP-PCSK9(156–227) showed a higher level of free PCSK9 than mice immunized with cVLP-PCSK9(FL) or naked SpyC-cVLP (Figure 5D). However, the level of IgG-bound PCSK9 was higher for mice immunized with cVLP-PCSK9(210–226) than in mice immunized with cVLP-PCSK9(156–227). This indicated that mice vaccinated with cVLP-PCSK9(210–226) had a larger proportion of PCSK9 bound in circulating immune complexes than mice vaccinated with cVLP-PCSK9(156–227) (Figure 5D,E). The total PCSK9 level in mice vaccinated with SpyC-cVLP (i.e., without PCSK9) was comparable to the pre-bleed levels (Figure 5C). Importantly, for these animals, there were no detectable levels of PCSK9 in the IgG-purified fraction, emphasizing the absence of PCSK9 immune complexes in this group (Figure 5E).

## 4. Discussion

It is well documented that the display of self-antigens on VLPs can enable the induction of antigen-specific auto-antibody responses in animals and humans [7,8,9,10,11] and that the quality of the antigen display is essential to effectively overcome B-cell tolerance [5,46,47,48]. In this study, we utilized our SpyT/SpyC AP205 cVLP vaccine technology to develop cVLP vaccines displaying either FL PCSK9- or PCSK9-derived peptide antigens, with the purpose to investigate potential differences in the capacity of vaccine-induced antibody responses to opsonize and clear plasma PCSK9. The shorter PCSK9 antigens include linear B-cell epitopes, previously shown to lower TC levels in both mice and macaques when displayed on the Qβ cVLP [34,35]. Initial quality assessment of all the cVLP-PCSK9 vaccines showed equally good vaccine stability, as well as the integrity of intact particles. The estimated coupling efficiency differed between the peptide-based vaccines (80/60%) and the cVLP-PCSK9(FL) (40%) vaccine, respectively. This is expected, as the larger FL PCSK9 antigen occupies more cVLP surface area than the smaller peptide antigens. However, the presence of excess antigen in each of the antigen coupling reactions indicates a complete surface decoration of cVLPs for all vaccines. Thus, potential variation in the density of the antigen display is not expected to influence the immunogenicity of the cVLP-PCSK9 vaccines. Accordingly, it was surprising to find that the initial quantitative analysis of vaccine-induced anti-PCSK9 antibody responses showed that mice vaccinated with the peptide-based PCSK9 cVLP vaccines had significantly higher anti-PCSK9 antibody levels than mice vaccinated with the cVLP-PCSK9(FL) vaccine. Specifically, since all the VLP-based vaccines looked similar in the quality assessment, we had expected the larger protein antigen to induce more anti-PCSK9 antibodies. Furthermore, when measuring plasma PCSK9, cVLP-PCSK9(FL)-vaccinated mice were found to have significantly lower levels compared to mice receiving either of the peptide-based PCSK9 cVLP vaccines. Thus, one explanation for these results could be that the comparatively broader anti-PCSK9 response, induced by the full-length antigen, resulted in more efficient opsonization and clearance of plasma PCSK9, resulting in decreased levels of both circulating PCSK9 and anti-PCSK9 antibodies. In support of this hypothesis, previous studies have reported significantly increased plasma PCSK9 levels in mice receiving anti-PCSK9 mAb or PCSK9 peptide-based vaccines (i.e., inducing narrow anti-PCSK9 Ab responses) [33,34,35,36,37,38,41]. In these cases, it was speculated that the binding of specific antibodies to PCSK9 does not effectively mediate immune clearance, but instead leads to the formation of circulating immune complexes with an extended half-life compared to free-circulating PCSK9 [33,34,35]. Moreover, it is expected that the induction of a broad polyclonal antibody response would lead to more efficient opsonization, complement fixation and subsequent immune clearance by phagocytes [49,50]. This general interpretation seems to agree with our additional results, which show that vaccination with either of the peptide-based PCSK9 cVLP vaccines leads to higher levels of IgG-bound PCSK9 than vaccination with the cVLP-PCSK9(FL) vaccine.

Collectively, our data indicate that induction of a broad polyclonal anti-PCSK9 antibody response (i.e., by the FL PCSK9 antigen) mediates increased opsonization and clearance of plasma PCSK9, compared to antibody responses induced by peptide-based antigens. Although, in this study, this effect did not significantly correlate with an increased capacity to lower plasma TC levels, this trait may still be important for the efficacy and safety of a vaccine. Furthermore, when developing vaccines against non-infectious diseases, it may be argued that the inclusion of FL self-antigens could increase the risk of activating harmful antigen-specific T cells [51]. Thus, although no apparent side effects were observed in this study, further work is needed to investigate whether immunization with the cVLP-PCSK9(FL) vaccine could lead to the activation of PCSK9-specific T cells in mice.

There are currently promising results being reported on several different PCSK9-inhibiting therapies, including other PCSK9 vaccines [34,35,36,37,38,39]. However, it can be difficult to directly compare results from such preclinical studies and deduce which one will be the most effective and safe in humans, especially for self-antigen-based vaccines that need to overcome B-cell tolerance. However, the recent approval by the FDA and EMA of the more cost-effective PCSK9-targeting RNAi therapy, Inclisiran [52,53,54,55,56], as well as other promising treatments (e.g., CRISPR gene-editing targeting PCSK9 [57,58]), may hamper clinical development of a PCSK9 vaccine. Nevertheless, a strong rationale remains for pursuing active vaccination (e.g., as an alternative to mAb therapy) against disease-associated self-antigen targets, and we believe that this study may be useful in guiding the design of other vaccines that should effectively mediate opsonization and immune clearance of the antigen target.

## 5. Conclusions

In this study, we performed a head-to-head immunization study to investigate whether induction of a broader polyclonal anti-PCSK9 antibody response would mediate more efficient clearance of plasma PCSK9. Specifically, we utilized the versatile SpyT/SpyC cVLP vaccine platform to deliver both FL- and peptide-based PCSK9 antigens for induction of vaccine-induced antibody responses of different breadth, as previous studies have shown that PCSK9-specific monoclonal antibodies result in circulating immune complexes. Our data show that mice vaccinated with a cVLP-PCSK9 FL vaccine show a higher capacity to opsonize and clear plasma PCSK9, as compared to mice vaccinated with peptide-based PCSK9 cVLP vaccines. These findings could help for future design of PCSK9/other vaccines to ensure high efficacy and safety of vaccines where target opsonization and immune clearance are desirable.

## Figures and Tables

**Figure 1 vaccines-11-00002-f001:**
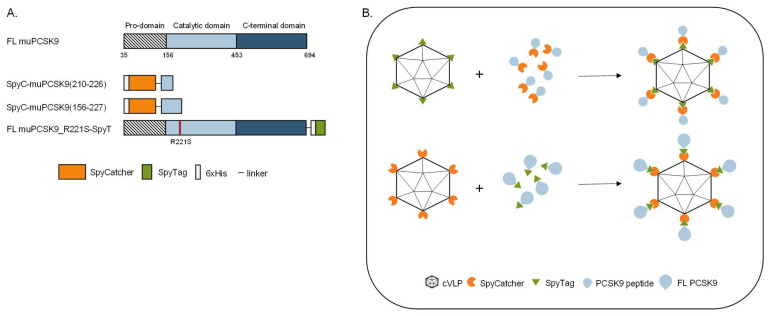
PCSK9 vaccine design. (**A**) Schematic representation of FL murine PCSK9 and the three PCSK9 antigen designs. FL muPCSK9 consists of a pro-domain, a catalytic domain and a C-terminal domain. Two peptide-based muPCSK9 antigens consisting of amino acid (aa) 210–226 and 156–227 were each N-terminally fused to a hexahistidine tag and SpyC, referred to as SpyC-PCSK9(210–226) and SpyC-PCSK9(156–227), respectively. The third antigen consists of the FL muPCSK9 construct, with the introduction of an R221S mutation to prevent furin cleavage, a C-terminal hexahistidine tag and SpyT, referred to as FL PCSK9-SpyT (**B**). Schematic representation of the Spy-AP205 technology used to create the cVLP-PCSK9 vaccines. Genetic fusion of either the SpyT (top) or SpyC (bottom) to the AP205 capsid protein (total of 180 subunits per cVLP) allows for unidirectional and high-density coupling of the PCSK9 antigen containing the corresponding binding partner.

**Figure 2 vaccines-11-00002-f002:**
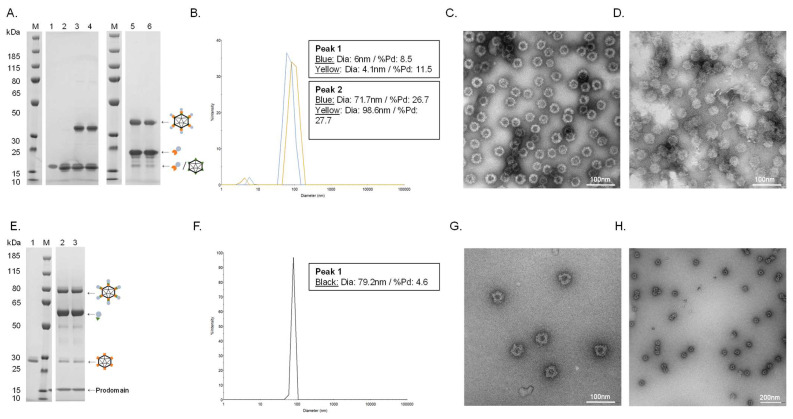
Vaccine quality assessment. (**A**). Individual vaccine component and resulting coupling band on a reduced SDS-PAGE gel. M: marker, lane 1: unconjugated SpyT-cVLP (16.5 kDa), lane 2: unconjugated SpyC-PCSK9(210–226) (15.5 kDa), lane 3: cVLP-PCSK9(210–226) after coupling overnight at 4 °C (32 kDa), lane 4: cVLP-PCSK9(210–226) after coupling overnight at 4 °C (32 kDa) + centrifugation test, lane 5: cVLP-PCSK9(156–227) after coupling overnight at 4 °C (38.5 kDa), lane 6: cVLP-PCSK9(156–227) after coupling overnight at 4 °C (38.5 kDa) + centrifugation test. (**B**). DLS analysis, showing a histogram of the % intensity of purified cVLP-PCSK9(210–226) (blue) and cVLP-PCSK9(156–227) (yellow) particles. The average diameter and polydispersity (%Pd) are annotated for each peak. (**C**). TEM image of negatively stained purified cVLP-PCSK9(210–226). (**D**). TEM image of negatively stained purified cVLP-PCSK9(156–227). (**E**). Individual vaccine component and resulting coupling band on a reduced SDS-PAGE gel. M: marker, lane 1: unconjugated SpyC-cVLP (27 kDa), lane 2: cVLP-PCSK9(FL) after coupling overnight at 4 °C (86 kDa), lane 3: cVLP-PCSK9(FL) after coupling overnight at 4 °C (86 kDa) + centrifugation test. (**F**). DLS analysis, showing a histogram of the % intensity of the purified cVLP-PCSK9(FL) particles. The average diameter and polydispersity (%Pd) are annotated. (**G**,**H**). TEM image of negatively stained purified cVLP-PCSK9(FL).

**Figure 3 vaccines-11-00002-f003:**
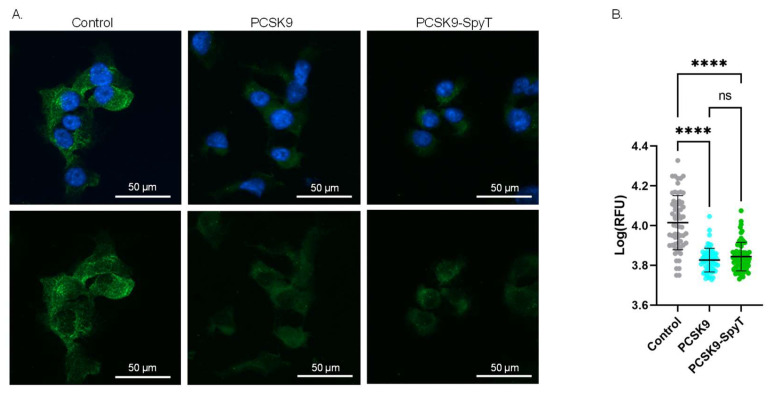
Biological activity of FL PCSK9-SpyT antigen by LDLR staining on Hepa1-6 cells. Hepa1-6 cells incubated with growth media (control), 100 nM FL muPCSK9 or 100 nM FL muPCSK9-SpyT for 4 h at 37 °C. Cells were stained with DAPI (blue) and anti-LDLR-FITC (green). (**A**). Representative cells are shown as merged pictures (top) or FITC signal alone (bottom). The size bar is 50 µm. (**B**). The relative fluorescence unit (RFU) was determined (*n* = 65) by Cytation5 software and depicted as mean ± SD. Statistical analysis was performed on log-transformed RFU values using one-way ANOVA, Tukey’s multiple comparisons test (adjusted *p*-value < 0.05 was accepted as significant, ns > 0.05, **** < 0.0001).

**Figure 4 vaccines-11-00002-f004:**
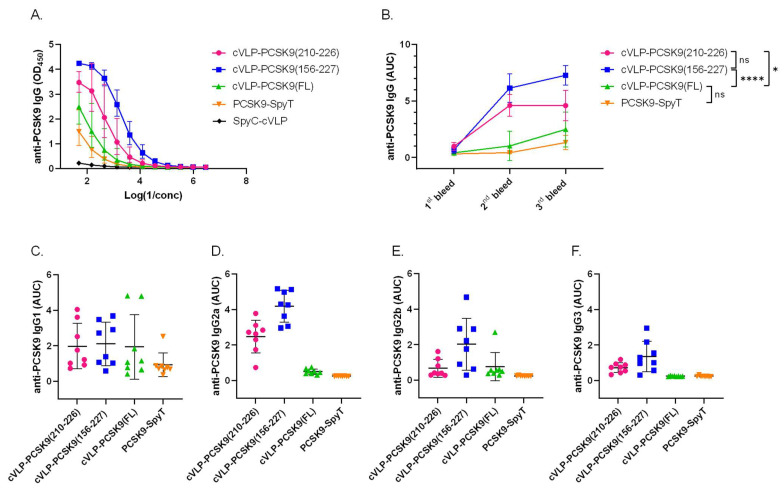
cVLP-PCSK9 vaccines induce antigen-specific antibody titers in mice. Plasma samples were obtained from male BALB/c mice (*n* = 8 per group) two weeks after prime (1st bleed), boost (2nd bleed) and second boost (3rd bleed), respectively. (**A**). Total IgG anti-PCSK9 dilution curve (3rd bleed) detected in mice vaccinated with cVLP-PCSK9(210–226) (pink), cVLP-PCSK9(156–227) (blue), cVLP-PCSK9(FL) (green), soluble PCSK9-SpyT (orange) or unconjugated SpyC-cVLP (black). Variance is shown as geometric mean ± geometric SD (**B**). Total IgG anti-PCSK9 titers are depicted as AUC with mean ± SD from 1st, 2nd and 3rd bleeds. (**C**–**F**). anti-PCSK9 IgG subclass (**C**). IgG1, (**D**). IgG2a, (**E**). IgG2b, (**F**). IgG3 depicted as AUC titer with mean ± SD detected in plasma from mice vaccinated with cVLP-PCSK9(210–226), cVLP-PCSK9(156–227), cVLP-PCSK9(FL) and soluble FL PCSK9-SpyT. Statistical analysis was performed on log-transformed AUC values using one-way ANOVA, Tukey’s multiple comparisons test (adjusted *p*-value < 0.05 was accepted as significant, ns > 0.05, * < 0.05, **** < 0.0001).

**Figure 5 vaccines-11-00002-f005:**
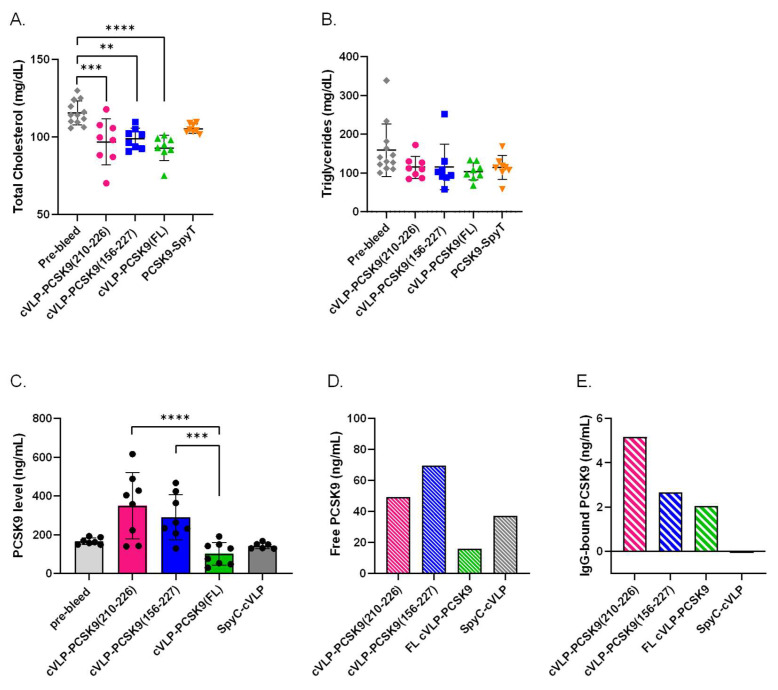
Biological effect of vaccine-induced antibodies. Plasma samples were obtained from groups (*n* = 8) of male BALB/c mice after three immunizations (3rd bleed) with cVLP-PCSK9(210–226), cVLP-PCSK9(156–227), cVLP-PCSK9(FL), PCSK9-SpyT or SpyC-cVLP. Data are depicted as the mean ± SD. (**A**). Total cholesterol (TC) level with mean ± SD measured in plasma from mice before vaccination (i.e., pre-bleed) (116 ± 8) or after vaccination with cVLP-PCSK9(210–226) (96.9 ± 15), cVLP-PCSK9(156–227) (98.9 ± 7), cVLP-PCSK9(FL) (92.9 ± 8) or soluble PCSK9-SpyT (105 ± 3). (**B**). Triglyceride level with mean ± SD detected in plasma obtained before vaccination (pre-bleed) (159 ± 67) or after vaccination with cVLP-PCSK8210-226) (115 ± 29), cVLP-PCSK9(156–227) (115 ± 59), cVLP-PCSK9(FL) (104 ± 22) or soluble PCSK9-SpyT (114 ± 31). (**C**). Quantitative total PCSK9 levels measured in plasma before vaccination (pre-bleed) (166 ± 18) or after vaccination with cVLP-PCSK9(210–226) (350 ± 170), cVLP-PCSK9(156–227) (290 ± 116), cVLP-PCSK9(FL) (102 ± 58.4) or SpyC-cVLP (143 ± 15.7). (**D**). Quantitative free PCSK9 levels detected in run through after purifying IgG from pooled plasma sample obtained from mice vaccinated with cVLP-PCSK9(210–226) (49.2 ng/mL), cVLP-PCSK9(156–227) (69.6 ng/mL), cVLP-PCSK9(FL) (16.1 ng/mL) or SpyC-cVLP (37.2 ng/mL). (**E**). Quantitative PCSK9 levels detected in 10 µg purified IgG from pooled plasma samples obtained from mice vaccinated with cVLP-PCSK9(210–226) (5.2 ng/mL), cVLP-PCSK9(156–227) (2.7 ng/mL), cVLP-PCSK9(FL) (2.1 ng/mL) or SpyC-cVLP (0.040 ng/mL). Statistical analysis was performed on log-transformed values using one-way ANOVA, Tukey’s multiple comparisons test and non-parametric, two-tailed, Mann–Whitney test (adjusted *p*-value < 0.05 was accepted as significant, ** < 0.05, *** < 0.001, **** < 0.0001).

## Data Availability

The data that support the findings of this study are available upon request. Accession codes are the following: murine PCSK9 protein (GenPept: NP_705793), SpyC-AP205 (Genbank: OK422508.1) and SpyTag-AP205 (Genbank: OK545878.1).

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
