# Peer review of "A cVLP-Based Vaccine Displaying Full-Length PCSK9 Elicits a Higher Reduction in Plasma PCSK9 Than Similar Peptide-Based cVLP Vaccines"

_vaccines, 2022, doi:10.3390/vaccines11010002_

Round 1

Reviewer 1 Report

Authors demonstrated the immunogenicity and antibody induction of capsid virus-like particle (cVLP)-carrying full-length (FL) PCSK9 and PCSK9-derived peptide antigens, respectively. They found a significantly increased capacity of the FL PCSK9 cVLP vaccine to clear plasma PCSK9. Capsid virus-like particle (cVLP) seems a good display platform, but the data among the immunized groups were controversial.  

Major comments:

1.       cVLP-PCSK9(156-227) induced higher titers of anti-PCSK9 than the others (Figure 4), but had a lower activity to clear plasma PCSK9 (Figure 5).

2.      If use the SpyC-cVLP or PCSK9-SpyT group as the control, there is no significance in the change of total cholesterol (TC) level in the groups of cVLP-PCSK9(210-226), cVLP- 421 PCSK9(156-227), and cVLP-PCSK9(FL) (Figure 5A).

Author Response

  1. cVLP-PCSK9(156-227) induced higher titers of anti-PCSK9 than the others (Figure 4), but had a lower activity to clear plasma PCSK9 (Figure 5).

Thank you for the insightful comment. Yes, the PCSK9-specific antibody levels are higher for mice vaccinated with cVLP-PCSK9(156-227), however as mentioned this does not correlate with increased clearance of circulating PCSK9. As seen in Figure 5C, the level of plasma PCSK9 is increased in mice vaccinated with either of the two peptide-based PCSK9-cVLP vaccines. But, where data in Figure 5D-E, indicate that mice vaccinated with the cVLP-PCSK9(210-226) vaccine have an increased level of PCSK9 bound in circulating immune complexes, the circulating PCSK9 seems to be more in the form of free soluble PCSK9 in mice vaccinated with the cVLP-PCSK9(156-227) vaccine – suggesting a decreased neutralizing activity of the vaccine-induced antibodies, which is also supported by the smallest reduction in TC levels by this vaccine, as compared to pre-bleed samples. To comment on this, we have included the following sentence in the discussion (line 480):

“By contrast, mice vaccinated with cVLP-PCSK9(156-227), showed relatively higher amount of free PCSK9 and less PCSK9 bound in immune complexes, suggesting that this vaccine despite the high levels of anti-PCSK9 antibodies elicited, had relatively less neutralization activity.”

  1. If use the SpyC-cVLP or PCSK9-SpyT group as the control, there is no significance in the change of total cholesterol (TC) level in the groups of cVLP-PCSK9(210-226), cVLP- 421 PCSK9(156-227), and cVLP-PCSK9(FL) (Figure 5A).

We thank the reviewer for the comment. It is correct that we did not comment on the differences in total cholesterol levels between mice immunized with full length PCSK9 in soluble form and displayed on cVLP. This difference was statistically significant, and this has been added to the result section (line 389):

“Moreover, cVLP display of FL PCSK9-SpyT results in a statistically significant reduction of TC (11.6%), as compared to mice vaccinated with soluble FL PCSK9-SpyT (p = 0.0002).“

We do not think that a comparison between vaccines using soluble full length PCSK9 and peptide-based cVLP vaccines are warranted as these vaccine employ different antigens.

Reviewer 2 Report

In this manuscript, Louise and colleagues utilized a versatile capsid virus-like particle (cVLP)-based vaccine platform to deliver both full-length PCSK9 and PCSK9 truncations to investigate whether induction of a broader polyclonal antibody response would mediate more efficient clearance of plasma PCSK9.  Overall this is an interesting study and significant data have been shown, while some of the experiment design may need improvements.

Specific comments:

1. line38, reference number cannot be used as object.

2.Fig.3, truncated and mutated PCSK9 images should also be included.

3.Fig.4, the control groups (SpyC-cVLP, Pre-bleed?) need to be shown in panel B-F.

Author Response

  1. line38, reference number cannot be used as object.

The sentence has been clarified and now states: “When mixed, SpyT and SpyC react to form a covalent binding through an isopeptide bond [6].”

  1. 3, truncated and mutated PCSK9 images should also be included.

Thank you for the comment. In Figure 3, a cell-based assay was used to confirm the biological activity of our full-length PCSK9-SpyTag antigen, by exploiting the ability to lower the level of the LDLR on hepatocytes. This assay was performed using the full-length PCSK9-SpyTag antigen (referred to by the referee as mutated) presented in Figure 1 to verify that the antigen we produced with a SpyTag was biologically active compared to a commercially available PCSK9. This was also the case when the antigen was displayed on a VLP (Supplementary Figure 2). Similar experiments were not performed with the truncated peptide-based PCSK9 antigens. These peptides do not span the LDLR binding site and are not expected to affect the LDLR level of hepatocytes.

  1. 4, the control groups (SpyC-cVLP, Pre-bleed?) need to be shown in panel B-F.

Thank you for the constructive comment. We did not show or investigate the anti-PCSK9 antibody levels in pre-vaccination samples. However, plasma from control mice vaccinated with the unconjugated cVLP, did not have a measurable anti-PCSK9 IgG response even after 3 immunizations (Figure 4A). Consequently, the anti-PCSK9 IgG subclass levels were not further investigated for this group.

Reviewer 3 Report

Well done.  Clearly presented and important results

Author Response

1. Well done.  Clearly presented and important results

Thank you very much.

Round 2

Reviewer 1 Report

The authors did not provide the new data to respond my comments, just amended the interpretation of the results and modified the disscision.

Author Response

1. cVLP-PCSK9(156-227) induced higher titers of anti-PCSK9 than the others (Figure 4), but had a lower activity to clear plasma PCSK9 (Figure 5).

We appreciate your comment and agree that, at first sight, the generated data could seem controversial. Specifically, we were indeed surprised to measure higher PCSK9-specific antibody levels in mice vaccinated with peptide-based vaccines (cVLP-PCSK9(156-227) and cVLP-PCSK9(210-226)), when comparing to mice immunized with the full-length PCSK9-cVLP. Since all the cVLP vaccines looked similar in the quality assessment, we had expected that the larger protein antigen induced more antibodies. However, after learning that vaccination with the FL-PCSK9 cVLP vaccine led to a significantly higher reduction in serum PCSK9 compared to vaccination with either of the peptide PCSK9 cVLP vaccines, we realized that the initial measured ELISA titres maybe have given the false impression that the FL-PCSK9 VLP vaccine induced the least antibodies. Instead, we rasied the hypothesis, that antibodies induced by the FL PCSK9-cVLP vaccines was more rapidly cleared from the circulation due to efficient opsonization of plasma PCSK9. This led us to examine quantitative differences in the amount of Free / IgG bound PCSK9 in mice from the different vaccination groups. This analysis showed that mice immunized with the peptide vaccines (especially the smallest peptide, PCSK9(156-227)) had higher levels of IgG-bound PCSK9 compared to mice vaccinated with FL-PCSK9 VLP. These results thus support the hypothesis that that the more polyclonal antibodies in FL-PCSK9 cVLP vaccinated mice, are more efficient at opsonizing and clearing plasma PCSK9 i.e. leading to a reduction in both PCSK9 and anti-PCSK9 antibodies.

To make these interpretations more clear, we have included the following text in the results (lines 402-410) and edited the discussion (line 463-485)): 

Results:

"Although, there was no statistically significant difference between the total PCSK9 level in pre-vaccination (pre-bleed) plasma and plasma obtained after immunization with either of the cVLP-PCSK9 vaccines, there was a clear trend that PCSK9 levels were higher in mice vaccinated with either of the peptide-based PCSK9 VLP vaccines (Fig. 5C). In fact, the level of total PCSK9 quantified in mice vaccinated with the peptide-based PCSK9 cVLP vaccines was statistically significantly higher than in mice vaccinated with cVLP-PCSK9 FL."

Discussion: 

"Accordingly, it was surprising to find that the initial quantitative analysis of vaccine-induced anti-PCSK9 antibody responses showed that mice vaccinated with the peptide-based PCSK9 cVLP vaccines had significantly higher anti-PCSK9 antibody levels than mice vaccinated with the cVLP-PCSK9(FL) vaccine. Specifically, since all the VLP-based vaccines looked similar in the quality assessment, we had expected the larger protein antigen to induce more anti-PCSK9 antibodies. Furthermore, when measuring plasma PCSK9, cVLP-PCSK9(FL) vaccinated mice were found to have significantly lower levels compared to mice receiving either of the peptide-based PCSK9 cVLP vaccines. Thus, one explanation for these results could be that the comparatively broader anti-PCSK9 response, induced by the full-length antigen, resulted in more efficient opsonization and clearance of plasma PCSK9, resulting in decreased levels of both circulating PCSK9 and anti-PCSK9 antibodies. In support of this hypothesis, previous studies have reported significantly increased plasma PCSK9 levels in mice receiving anti-PCSK9 mAb or PCSK9 peptide-based vaccines (i.e. inducing narrow anti-PCSK9 Ab responses) [33–38,41]. In these cases, it was speculated that the binding of specific antibodies to PCSK9 does not effectively mediate immune clearance, but instead leads to the formation of circulating immune complexes with an extended half-life compared to free circulating PCSK9 [33–35]. Moreover, it is expected that the induction of a broad polyclonal antibody response leads to more efficient opsonization, complement fixation, and subsequent immune clearance by phagocytes [49,50]. This general interpretation seems to agree with our additional results, which show that vaccination with either of the peptide-based PCSK9 cVLP vaccines leads to higher levels of IgG-bound PCSK9 than vaccination with the cVLP-PCSK9(FL) vaccine."

2. If use the SpyC-cVLP or PCSK9-SpyT group as the control, there is no significance in the change of total cholesterol (TC) level in the groups of cVLP-PCSK9(210-226), cVLP- 421 PCSK9(156-227), and cVLP-PCSK9(FL) (Figure 5A).

We thank the reviewer for the comment. It is correct that we did not comment on the differences in total cholesterol levels between mice immunized with full length PCSK9 in soluble form and displayed on cVLP. This difference was statistically significant, and this has been added to the result section (line 389):

“Moreover, cVLP display of FL PCSK9-SpyT results in a statistically significant reduction of TC (11.6%), as compared to mice vaccinated with soluble FL PCSK9-SpyT (p = 0.0002).“

We do not think that a comparison between vaccines using soluble full length PCSK9 and peptide-based cVLP vaccines are warranted as these vaccine employ different antigens.

Reviewer 2 Report

the authors' responses released my concerns, I have no more questions

Author Response

Thank you.